# Privacy in Community Pharmacies in Saudi Arabia: A Cross-Sectional Study

**DOI:** 10.3390/healthcare12171740

**Published:** 2024-08-31

**Authors:** Marwan A. Alrasheed, Basmah H. Alfageh, Omar A. Almohammed

**Affiliations:** 1Department of Clinical Pharmacy, College of Pharmacy, King Saud University, Riyadh 11451, Saudi Arabia; balfajeh@ksu.edu.sa (B.H.A.); oalmohammed@ksu.edu.sa (O.A.A.); 2Pharmacoeconomics Research Unit, College of Pharmacy, King Saud University, Riyadh 11451, Saudi Arabia

**Keywords:** patient privacy, community pharmacists, pharmacy design, healthcare services, patient–pharmacy interactions

## Abstract

Background: Privacy in healthcare is a fundamental right essential to maintain patient confidentiality and trust. Community pharmacies in Saudi Arabia (SA) play a critical role in the healthcare system by providing accessible services and serving as initial points of contact for medical advice. However, the open nature of these settings poses significant challenges in maintaining patient privacy. Methods: This cross-sectional study used electronic surveys distributed across various online platforms. The target sample included Saudi adults, with a sample size of 385 participants to achieve 80% statistical power at a 95% confidence interval. The survey comprised demographic questions and sections evaluating perceptions of privacy, the importance of privacy, and personal experiences regarding privacy in community pharmacies. Descriptive statistics and logistic regression models were used for the analysis. Results: A total of 511 responses were obtained. The mean age was 33.5 years, with an almost equal distribution of males (49.71%) and females (50.29%). Most participants held a bachelor’s degree or higher (78.67%). Privacy perceptions varied, with only 9.0% strongly agreeing that there was a private space for consultations, while 64.0% felt that the design of community pharmacies did not adequately consider patient privacy, and 86.9% reported that conversations could be overheard. Privacy concerns were notable, with almost one-half of the participants (47.6%) having concerns about privacy and 56.6% doubting the confidentiality of their health information. Moreover, 17.6% reported being asked for unnecessary personal information when buying medication, and 56.2% admitted to avoiding discussing a health problem with the pharmacist due to privacy concerns. Experiences of privacy breaches were reported by 15.7% of respondents. Logistic regression analysis revealed that the availability of private space in the pharmacy and patients feeling that the pharmacy respects their privacy were associated with a lower likelihood of avoiding discussions with pharmacists due to privacy concerns (OR = 0.758, CI = 0.599–0.0957 and OR = 0.715, CI = 0.542–0.945 respectively) Conversely, greater privacy concerns and previous privacy breaches significantly increased the likelihood of avoiding discussions with pharmacists in the community pharmacy (OR = 1.657, CI = 1.317–2.102 and OR = 4.127, CI = 1.886–9.821 respectively). Conclusions: This study highlights the significant concerns regarding privacy practices in community pharmacies in SA. Thus, there is a need for standards to improve privacy in community pharmacies, such as mandating the need for private consultation areas and enhanced staff training on handling privacy-related issues. Addressing the issue of privacy is crucial for maintaining patient trust, improving healthcare service quality, and ensuring effective patient–pharmacist interactions.

## 1. Introduction

Healthcare privacy is essential for maintaining patient confidentiality and trust. Ensuring privacy is critical in community pharmacies where sensitive health information is frequently exchanged, and ensuring privacy is particularly critical [1,2,3]. Community pharmacies are integral to the healthcare system and often serve as the initial point of contact for patients seeking medical advice and prescription services [4]. However, the open and accessible nature of these settings can present significant challenges to maintaining patient privacy [5]. Ensuring that patient information is protected from unauthorized access and interactions with pharmacists are conducted confidentially is crucial for safeguarding patient trust and upholding professional ethical standards [6].

Privacy was first defined in 1890 as the right to be left alone [7]. Several decades later, particularly in 1948, the right to privacy was one of the twelfth fundamental rights approved in the Universal Declaration of Human Rights [8]. In 1981, the Council of Europe approved the Data Protection Convention (Convention 108); this was the first time that the right to privacy was protected into the European law [9]. Later, in 2018, the United Nations enacted the Personal Data Protection and Privacy Principles as the primary source for the protection of personal data by all United Nations institutions [10]. Privacy in the healthcare system was defined as the right and desire of a person to control the disclosure of personal health information [11]. According to American Medical Association (AMA) principles of medical ethics, privacy in healthcare comprises several aspects: personal space (physical privacy), personal data (informational privacy), personal choices, including cultural and religious affiliations (decisional privacy), and personal relationships with family members and other intimates (associational privacy) [12]. The core principles of data protection in the context of public health include the following: fair, lawful and transparent, purpose limitation, accuracy, data minimization, storage minimization, rights of data subjects, integrity and confidentiality, and the international transfer of personal data [13].

Community pharmacies in Saudi Arabia (SA) are vital to the healthcare system, offering accessible services, such as medication dispensing, health advice, and preventive care. Their number has grown significantly over the years, highlighting their role in reforming the Saudi healthcare system [14]. They provide essential services, such as smoking cessation support and chronic disease management [15]. However, challenges remain, such as the need for better integration and patient-centered care practices [16]. The Saudi Ministry of Health has been actively working to enhance the role of community pharmacies through various initiatives aimed at improving the quality of pharmaceutical care and ensuring the implementation of best practices in medication management and patient counseling [17,18]. Despite these efforts, challenges remain in ensuring adequate privacy protection given the busy and open nature of these environments. While there is general awareness among pharmacy staff about the importance of patient privacy, there is a need for more structured training and clear guidelines to standardize privacy practices across all community pharmacies [16].

The Health Insurance Portability and Accountability Act (HIPAA) in the United States set a precedent for privacy standards in healthcare systems. HIPAA’s Privacy Rule establishes national standards to protect individuals’ medical records and other personal health information, thereby providing patients with significant rights to their health information [19,20]. This included the right to examine and obtain copies of health records and request corrections. The HIPAA Privacy Rule mandates appropriate safeguards to protect the privacy of personal health information and sets limits on the use and disclosure of such information without patient authorization. Similarly, in the United Kingdom (UK), the General Pharmaceutical Council (GPhC) mandates that all pharmacies must have consultation rooms to ensure private and confidential discussions between patients and pharmacists [21]. These regulations serve as valuable benchmarks for assessing and improving privacy practices in community pharmacies globally [20,22]. In 2018, the European Union (EU) implemented the General Data Protection Regulation (GDPR) to replace its previous legal framework on personal information privacy [23]. While both HIPAA and GDPR aim to protect data privacy, key differences exist. HIPAA specifically covers health-related data, known as Protected Health Information (PHI), and applies to entities handling PHI, such as health plans and providers. In contrast, GDPR has a broader scope, covering all personal data of EU citizens, regardless of the entity’s location. Both regulations grant individuals the rights to access, correct, and know how their data are used. However, GDPR requires explicit consent for data processing, which can be withdrawn at any time, and includes the right to erasure, allowing individuals to request the deletion of their data under certain conditions. HIPAA, on the other hand, requires medical records to be maintained for a specified period, even if the individual no longer interacts with the covered entity [24].

Recent studies have increasingly emphasized the critical importance of privacy in community pharmacies. Inadequate privacy in these settings can erode patient trust and hinder open communication, both of which are essential for effective healthcare delivery. Research indicates that privacy concerns significantly impact the utilization of pharmacy services globally, with patients often expressing dissatisfaction due to the lack of private consultation spaces. For example, a study in Iran found that crowded pharmacies compromise patient privacy, leading to lower satisfaction levels [25]. Similarly, a survey in Saudi Arabia highlighted that the absence of privacy is a major barrier to effective patient–pharmacist interactions [26]. These findings underscore the urgent need for community pharmacies to address privacy concerns to maintain patient trust and improve healthcare outcomes.

Moreover, implementing dedicated consultation areas and enhancing staff training have proven effective in improving patient satisfaction and overall care quality. A study in Malaysia, for instance, highlighted the necessity of upgrading privacy protection in community pharmacies, aligning with guidelines that advocate for private consultation spaces [27]. The presence of such areas not only fosters a more comfortable environment for patients but also enhances the quality of interactions between pharmacists and patients, thereby reinforcing trust [28]. Furthermore, research suggests that when patients feel secure in their privacy, they are more likely to engage openly with pharmacy staff, which is crucial for effective medication management and counseling [29,30]. This relationship between privacy and trust highlights the importance of creating environments where patients feel safe to discuss sensitive health issues.

The gap in the existing literature that this study addresses is the insufficient exploration of privacy concerns within community pharmacies in Saudi Arabia, particularly from the perspective of patients. While previous studies have highlighted privacy issues in various global contexts, there is a lack of focused research on how these concerns specifically impact patient trust, communication, and the overall effectiveness of healthcare delivery in Saudi community pharmacies. In this study, we investigated the current state of privacy practices in community pharmacies across SA, focusing on customer perceptions and experiences. The objective of this study was to evaluate customer perceptions of privacy practices in community pharmacies, assess the adequacy of existing privacy-protection measures, and identify potential gaps from customers’ perspectives. Thus, the study seeks to present a clear depiction of the current state of privacy in community pharmacies in SA and offer recommendations for improvements. By examining these aspects, we aimed to provide insights that can assist policymakers in improving privacy practices in community pharmacies, ultimately enhancing the overall quality of healthcare services in SA. By addressing these issues, we hope to contribute to the enhancement of patient confidentiality and overall effectiveness of pharmacy services in SA.

## 2. Methodology

### 2.1. Study Design, Participants, Setting, and Ethical Consideration

This was a cross-sectional study conducted in SA using a questionnaire that was constructed by the authors and disseminated through different online platforms to understand the privacy concerns in community pharmacies among the Saudi population. The sample comprised adult Saudi citizens and residents of the general population. Ethical approval was obtained from the Standing Committee for Scientific Research Ethics at King Saud University (reference no. KSU-HE-23-1166). The participants electronically signed a consent form prior to participating in the study. The collected data did not contain any information related to the person’s identity, and all gathered information was treated confidentially for research purposes.

### 2.2. Data Collection, Data Source, and Study Variables

Data were collected using an online survey. The survey questions were divided into two main sections. The first section gathered demographic data about the participants, such as age, gender, educational level, place of residence, and frequency of visits to community pharmacies. The second section was divided into three subsections to understand the participants’ perceptions of privacy in community pharmacies. The first subsection was composed of multiple questions with an agreement Likert scale to evaluate the participants’ opinions about privacy in community pharmacies and the adequacy of existing privacy-protection measures. The second subsection comprised two questions assessing the importance of privacy in community pharmacies to participants and their privacy concerns. The last subsection of the survey questions explored the participants’ privacy experiences in community pharmacy settings (Appendix A). To ensure content validity, a multi-step approach was implemented. Initially, a comprehensive literature review was conducted to identify all relevant aspects of the concept, though it was noted that the literature lacks detailed information specific to the situation in Saudi Arabia. The questionnaire was then subjected to expert evaluation by three specialists in the field—two assistant professors and one associate professor with expertise in health outcomes and pharmacoepidemiology. Their feedback was instrumental in confirming that the questionnaire comprehensively covered the topic. Additionally, a pilot study was conducted with six participants from the target population to assess the clarity and relevance of the questions. Based on the insights gained, necessary adjustments were made to enhance the content validity of the questionnaire. 

### 2.3. Statistical Analysis

Descriptive statistics, such as means and standard deviations, were used to present the results of continuous variables, such as age, while percentages and frequencies were used to present categorical variables, such as sex, educational level, and degree of agreement with the questions. In the analysis, the responses from the Likert scale questions were merged into three categories: “Strongly agree or agree”, “Neutral”, and “Strongly disagree or disagree”. This approach was applied to simplify the presentation and interpretation of the data. A logistic regression model was employed to predict the potential reasons behind avoiding discussing a health problem with pharmacists due to privacy concerns and to predict the potential factors linked to a previous experience of a privacy breach in community pharmacies, with odds ratios (ORs) and 95% confidence intervals (95% CIs) included. Cronbach’s alpha was used to assess the internal consistency of the questions. The target sample size to provide 80.0% statistical power of the study results, with a 95% confidence interval, was 385 participants. The sample size calculation was performed using G*Power 3.1 software to ensure an adequate number of participants for a reliable statistical analysis [31]. Other statistical analysis in this study was conducted using R statistical software version 4.2.2 [32].

## 3. Results

### 3.1. Demographic Characteristics

In total, 511 responses were obtained, and the mean age for the participants was 33.5 years (±12 years), with an almost equal distribution of males (49.7%) and females (50.3%). However, the educational level varied among the participants, with the majority holding a bachelor’s (46.2%) or higher (32.5%) educational degree. Regarding employment status, 58.7% were employed, while 39.1% were unemployed. Approximately one-half of the participants were married (51.7%), and 3.5% were divorced or widowed. Regarding the frequency of community pharmacy visits, 60.3% visited the community pharmacies once a month or less, while 32.1% visited them 2–3 times a month. The Cronbach’s alpha coefficient for the items was 0.79. The demographic characteristics of the study participants are summarized in Table 1.

### 3.2. Privacy Perceptions and Practices in Community Pharmacies

The survey results revealed mixed perceptions of privacy measures in community pharmacies. About one-third of the participants (30.9%) strongly agreed or agreed that there was a private space for consultation in community pharmacies, while one-half of them (50.1%) disagreed or strongly disagreed. Regarding the statement about the design of pharmacies being considerate for privacy, only 17.4% agreed, while 64.0% disagreed. Moreover, only 20.5% of the participants agreed that customers had sufficient personal space when talking to a pharmacist, while 26.6% felt that the place respects their privacy when consulting pharmacists (Figure 1).

Most participants (86.9%) agreed that conversations between pharmacists and patients could be easily overheard by other customers in the community pharmacy. Regarding pharmacists’ professionalism, 59.4% agreed that pharmacists act professionally when discussing their health information, while only 12.2% disagreed with this statement. Only 16.8% strongly agreed or agreed that the environment was suitable for private discussions, whereas 62.8% disagreed with that. In terms of whether pharmacies educate customers about privacy, only 13.3% strongly agreed or agreed, while 66.5% disagreed or strongly disagreed (Figure 1).

### 3.3. Importance of Privacy and Privacy Concerns in Community Pharmacies

Most respondents (82.1%) rated the importance of having a private space in community pharmacies as very important or important. However, the respondents’ concerns about privacy were significant, with 19.4% being very concerned and 28.2% being somewhat concerned, while a substantial portion (35.2%) remained neutral and 17.3% were not concerned (Figure 2A,B).

### 3.4. Patient Experiences with Privacy in Community Pharmacies

The survey addressed specific privacy practices and experiences of community pharmacies. A large majority (82.4%) reported never having been asked for unnecessary personal information, whereas 17.6% had experienced it. More than one-half of the participants (56.2%) avoided discussing their health problems with a pharmacist because of privacy concerns, whereas 43.8% did not. Regarding the confidentiality of their health information, 43.4% believed it was treated confidentially, whereas 56.6% did not. Additionally, a significant majority (84.3%) had no previous experience of privacy breaches; however, 15.7% reported such an experience (Figure 3).

### 3.5. Predictors of Privacy Concerns and Breaches in Community Pharmacies

A logistic regression model predicting whether the participants avoided discussing a health problem with a pharmacist because of privacy concerns revealed several significant predictors. The availability of private space to talk to the pharmacist (*β* = −0.313, OR = 0.758, *p*-value = 0.020), feeling that the pharmacy respects privacy (*β* = −0.534, OR = 0.715, *p*-value = 0.018), and having a suitable environment to discuss private health information (*β* = −0.353, OR = 0.703, *p*-value = 0.045) decreased the likelihood of avoiding discussions. Contrariwise, higher levels of privacy concern (*β* = 0.509, OR = 1.657, *p*-value < 0.001) and having past experiences of privacy breaches (*β* = 1.418, OR = 4.127, *p*-value < 0.001) significantly increased the likelihood of avoiding discussions. Believing that personal health information is treated confidentially (*β* = −0.625, OR = 0.473, *p*-value = 0.001) was associated with a decreased likelihood of avoiding discussions (Table 2).

Further, the logistic regression model examined whether the participants experienced a privacy breach during their visit to a community pharmacy setting and found that gender is a significant predictor, with males being more likely to report a privacy breach (*β* = 0.735, OR = 2.085, *p* = 0.020). Having a private space to talk to the pharmacist (*β* = −0.442, OR = 0.643, *p*-value = 0.015) and the perception that pharmacies adequately educate customers on patient privacy (β = −0.485, OR = 0.615, *p* = 0.032) were associated with a lower likelihood of a privacy breach. Avoiding discussions with pharmacists due to privacy concerns (*β* = 1.456, OR = 4.291, *p*-value < 0.01) and being asked for unnecessary personal information (*β* = 1.697, OR = 5.458, *p*-value = 0.01) were significantly associated with a higher likelihood of experiencing a privacy breach. These results highlighted the key factors associated with privacy breaches in community pharmacies (Table 3).

## 4. Discussion

This study highlights significant concerns about privacy practices in community pharmacies across SA, thereby revealing critical areas for improvement. The findings indicate a substantial lack of private consultation spaces, with over half of the respondents (50.1%) disagreeing or strongly disagreeing that such spaces exist, which impedes open patient–pharmacist communication. Furthermore, 64.0% of participants felt that the design of community pharmacies did not adequately consider patient privacy, suggesting the need for structural redesigns to enhance privacy. Patient perceptions of respect for their privacy were notably low, with only 26.6% feeling respected during their interactions and 86.9% reporting that conversations could be overheard. While some acknowledged the professionalism of pharmacists, there was a clear need for better privacy education, as only 13.3% felt sufficiently informed about the importance of privacy practices. Privacy concerns were significant, with 47.5% of the respondents expressing high levels of concern and 56.6% doubting the confidentiality of their health information. Experiences of privacy breaches, reported by 15.7% of the respondents, notably influenced patient behavior, increasing the likelihood of avoiding health discussions with pharmacists.

Community pharmacy customers in SA have been concerned about privacy for the past two decades [33]. This concern is shared among the community pharmacy customers worldwide [34]. The lack of privacy protection negatively impacts healthcare outcomes. It has been reported as a key barrier to the use of extended community pharmacy services by 58% of participants in a cross-sectional study conducted in SA [35]. In Australia and UK, although community pharmacies are required by law to have an area where consultations cannot reasonably be expected to be overheard, privacy is counted as a barrier to information exchange during over-the-counter (OTC) medication consultations in community pharmacies [21,36,37].

A cross-sectional study conducted in Madinah, a city in the western province of SA, aimed to explore the level and quality of privacy offered by community pharmacies, using simulated client surveys in 80 random community pharmacies [26]. This study found that physical measures of privacy, such as designated consultation areas, were available in only one pharmacy out of the 80 pharmacies, and they were not used and not designed separately from the pharmacist’s counter. Furthermore, conversations during the simulated customer–pharmacist consultation were audible to the simulated customer in 63% of the pharmacies. The findings of this study were consistent with our participants’ perceptions.

In this study, a significant proportion of the patients (56.2%) reported avoiding discussions with their pharmacists about health problems because of privacy concerns. This avoidance can have profound implications on the patient’s health and well-being. Effective communication between patients and pharmacists is essential for the accurate dispensation of medications and the provision of vital health advice. When patients withhold this information, it can lead to suboptimal treatment outcomes, medication errors, and poor overall health. The pharmacist–patient relationship is fundamentally built on trust and confidentiality. Any breach or perceived lack of privacy can severely undermine this trust, ultimately affecting the quality of care that patients receive. Addressing privacy concerns through improved pharmacy design, such as by creating private consultation areas and employing sound barriers, is crucial. Additionally, ensuring that staff are well-trained to maintain confidentiality and clear communication protocols can help foster an environment in which patients feel safe and are more willing to share sensitive health information necessary for their treatment. This approach is essential to ensure patients receive comprehensive and effective care, highlighting the importance of prioritizing privacy in community pharmacies [38,39,40].

The utilization of community pharmacy spaces is one of the measures that has been applied to enhance privacy. This can be achieved by having an enclosed private area or by using visual and sound barriers to create visual and auditory privacy. Various additional strategies have been adopted by pharmacy staff in Australia to overcome privacy obstacles, such as taking consumers to a quieter part of the pharmacy, avoiding exposure to sensitive items through packaging, lowering voices, interacting during pharmacy quiet times, and telephoning consumers [41,42].

The findings of this study underscore the need for enhanced privacy measures in community pharmacies, including the implementation of private consultation areas, redesigning pharmacy layouts, and improving staff training on privacy issues. Establishing regulatory standards similar to those of the HIPAA, alongside mandatory training programs and patient education initiatives, could significantly enhance privacy practices in Saudi community pharmacies. Addressing these issues is crucial for maintaining patient trust, improving healthcare service quality, and ensuring effective patient–pharmacist interactions.

The findings from this study indicate that patients’ perceptions of pharmacy design, specifically their belief that the design takes their privacy into account, significantly influence their behavior. Notably, patients who perceived the pharmacy design as privacy-conscious were substantially more likely to avoid discussions with pharmacists (OR = 1.429). This suggests that a heightened awareness of privacy considerations might lead patients to become more cautious, resulting in the avoidance of potentially sensitive conversations. Additionally, the study revealed that patients who believed that the pharmacy design was focused on privacy were also more likely to report having experienced a privacy breach (OR = 1.833). This may suggest that when patients perceive a strong emphasis on privacy, it increases their sensitivity to privacy issues, making them more likely to perceive or recall breaches. These findings highlight the need to carefully balance patient perceptions with physical design elements to prevent unintended increases in privacy-related concerns.

Although the full impact of this study remains to be fully explored, implementing improved privacy measures is likely to enhance patient satisfaction and trust, encouraging open communication between patients and pharmacists and ultimately leading to better health outcomes. Regulatory bodies and pharmacy management should consider these findings to refine policies and practices that prioritize patient privacy. The reliability of the newly created scale was assessed using Cronbach’s alpha, which yielded a coefficient of 0.79. This indicated acceptable internal consistency, consistent with the standards suggested by Tavakol and Dennick (2011) [43]. An alpha value of 0.79 suggests that the items measure the underlying construct reliably, enhancing the credibility of our findings. This level of reliability supports the robustness of our data and ensures the dependability of the scale for assessing patient privacy. Future studies should validate the scale in diverse populations to confirm its utility.

The results of the logistic regression analysis revealed some small beta coefficients, which, while statistically significant, indicate a modest effect size. This suggests that the predictor variable in question exerts a limited influence on the outcome variable. Although the effect is small, it is important to consider the potential cumulative impact in contexts involving large populations or multiple contributing factors. Even slight shifts in probability may hold practical significance when scaled appropriately. Therefore, while the statistical significance of the findings is clear, the practical implications should be interpreted with caution, particularly in relation to the broader applicability of these results in clinical or policy decision-making.

This comprehensive cross-sectional study with a large sample size provides a depiction of current privacy practices in Saudi community pharmacies. However, this study had several limitations. Reliance on self-reported data may introduce response bias, and the study’s cross-sectional nature precludes causal inferences. Additionally, the sample may not fully represent all the regions of Saudi Arabia, thereby limiting the generalizability of the findings. Further research is warranted to explore the implementation of these recommendations and their effects on patient outcomes. Moreover, one of the limitations of this study is the potential skewness in the sample demographics, which may affect the representativeness of the findings. While the sample provides valuable insights, the demographic distribution may not fully reflect the broader population. This limitation should be considered when interpreting the results, and future research should aim to include a more diverse and representative sample to enhance the generalizability of the findings. Also, the statistical power of 80% was based on random sampling assumptions. However, convenience sampling via social media, resembling snowball sampling, could introduce biases, affecting sample representativeness and potentially reducing the study’s effective power. This may limit the generalizability of the findings, as the sample might not fully reflect the broader population. To improve future studies, more rigorous methods, like stratified random sampling or targeted recruitment, could enhance both the robustness and generalizability. Another limitation of this study is the inability to apply post-stratification weighting techniques to adjust for demographic imbalances in the sample. The accurate application of these techniques requires detailed population-distribution data specific to the Saudi Arabian context, which were not available. As a result, potential biases arising from the overrepresentation of certain demographic groups could not be corrected. It is suggested that post-stratification weighting be considered in future research if comprehensive population data become accessible.

## 5. Conclusions

Patient privacy is vital for maintaining patients’ trust in healthcare systems. The lack of privacy protection could obstruct the extended role of healthcare providers, which has a negative impact on healthcare outcomes. Although certain measures are implemented in community pharmacy settings to ensure patient confidentiality, from the patient’s perspective, such measures are inadequate, and several concerns about their privacy have been raised. Future in-depth studies that assess these concerns are required to provide optimal solutions and improve privacy-protection measures in community pharmacies.

## Figures and Tables

**Figure 1 healthcare-12-01740-f001:**
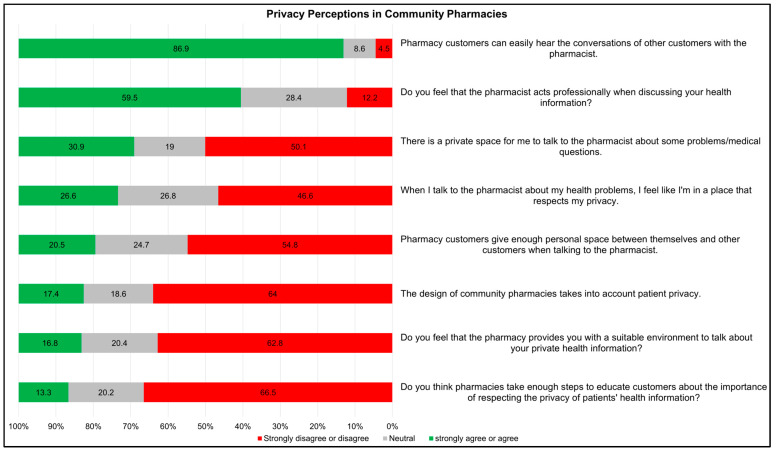
Privacy perceptions in community pharmacies.

**Figure 2 healthcare-12-01740-f002:**
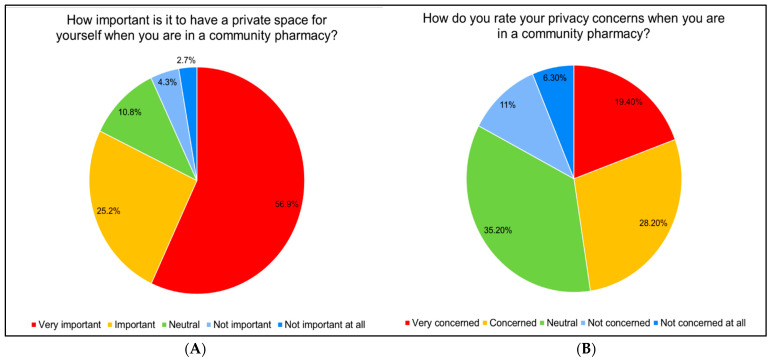
Importance of privacy and privacy concerns in community pharmacies. (**A**) The percentage distribution of responses to the question “How important is it to have a private space for yourself when you are in a community pharmacy?”. (**B**) The percentage distribution of responses to the question “How do you rate your privacy concerns when you are in a community pharmacy?”.

**Figure 3 healthcare-12-01740-f003:**
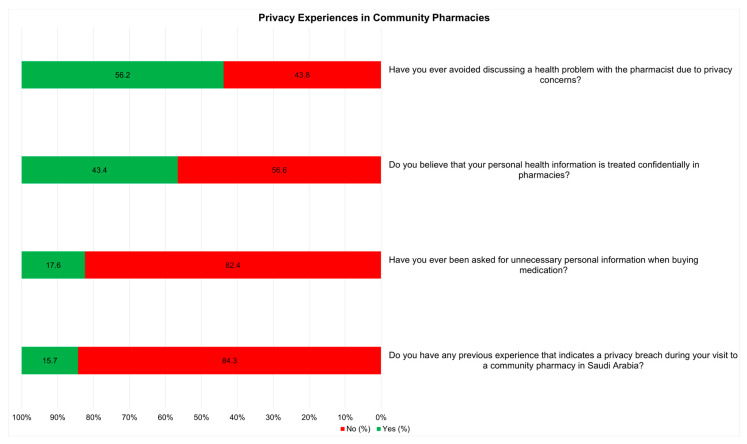
Privacy experiences in community pharmacies.

**Table 1 healthcare-12-01740-t001:** Demographic characteristics and community pharmacy visit frequency for study participants.

Variable	Mean ± SD or Frequency (%)
**Age**	33.5 ± 12
**Sex**	
Male	254 (49.7)
Female	257 (50.3)
**Educational level**	
Elementary School	3 (0.6)
Middle school	13 (2.5)
High School	93 (18.2)
Bachelor’s	236 (46.2)
Higher Education	166 (32.5)
**Occupational status**	
Employed	300 (58.7)
Not employed	200 (39.1)
Retired	11 (2.2)
**Marital Status**	
Single	229 (44.8)
Married	264 (51.7)
Divorced	14 (2.7)
Widowed	4 (0.8)
**Community pharmacy visits per month**	
Once every month or less	308 (60.3)
2–3 times a month	164 (32.1)
4 times or more a month	39 (7.6)

Numbers are presented as the mean ± SD or frequency (%).

**Table 2 healthcare-12-01740-t002:** Logistic regression of avoiding discussion with the pharmacist due to privacy concerns.

Variable	*β*	Std. Error	Odds Ratio	95% CI	z Value	*p*-Value
Age	−0.007	0.014	0.993	0.966–1.020	−0.53	0.596
Sex	−0.420	0.234	0.657	0.414–1.036	−1.796	0.072
Educational level	−0.048	0.104	0.924	0.660–1.293	−0.464	0.643
Occupational status	0.400	0.300	1.492	0.831–2.703	1.333	0.183
Marital status	−0.085	0.250	0.919	0.560–1.498	−0.34	0.733
Frequency of community pharmacy visits	0.005	0.173	1.005	0.709–1.428	0.031	0.976
Private space availability	−0.313	0.133	0.758	0.599–0.957	−2.327	0.020 *
Pharmacy design	0.359	0.166	1.429	1.036–1.988	−2.152	0.031 *
Importance of privacy	0.117	0.168	1.163	0.925–1.464	1.289	0.197
Customers give enough privacy	−0.275	0.126	0.875	0.683–1.121	−1.058	0.290
Respect privacy when speaking with a pharmacist	−0.534	0.225	0.715	0.542–0.945	−2.368	0.018 *
Customers can overhear conversations	0.146	0.146	1.278	0.962–1.704	1.687	0.092
Pharmacist professionalism	0.160	0.158	1.150	0.878–1.510	−1.014	0.310
Good environment for privacy	−0.353	0.176	0.703	0.496–0.988	−2.009	0.045 *
Pharmacies adequately educate customers on patient privacy	0.080	0.153	1.232	0.916–1.668	0.527	0.598
Level of worry about privacy	0.509	0.119	1.657	1.317–2.102	4.269	<0.01 *
Asked for unnecessary personal information	0.333	0.226	1.636	0.859–3.187	1.476	0.140
Health information treated confidentially	−0.625	0.234	0.473	0.298–0.748	−3.196	<0.01 *
Previous experience of privacy breach	1.418	0.418	4.127	1.886–9.821	3.388	<0.01 *

* *p*-value < 0.05.

**Table 3 healthcare-12-01740-t003:** Logistic regression of previous experience of a privacy breach in community pharmacies.

Variable	*β*	Std. Error	Odds Ratio	95% CI	z Value	*p*-Value
Age	−0.017	0.019	0.983	0.945–1.020	−0.901	0.368
Sex	0.735	0.316	2.086	1.133–3.927	2.327	0.020 *
Education level	0.432	0.243	1.540	0.966–2.507	1.780	0.075
Job status	−0.541	0.464	0.578	0.232–1.437	0.973	0.331
Marital status	0.072	0.336	1.075	0.551–2.063	0.215	0.830
Frequency of community pharmacy visits	0.312	0.429	1.394	0.915–2.110	0.563	0.103
Private space availability	−0.442	0.181	0.643	0.444–0.907	−2.434	0.015 *
Pharmacy design	0.602	0.211	1.833	1.174–2.907	2.849	<0.01 *
Importance of privacy	−0.038	0.170	0.963	0.697–1.359	−0.224	0.823
Customers give enough privacy	−0.054	0.198	0.767	0.535–1.087	−0.270	0.787
Respect privacy when speaking with a pharmacist	−0.180	0.190	0.835	0.569–1.206	−0.944	0.345
Customers can overhear conversations	0.154	0.162	1.166	0.786–1.768	0.947	0.344
Pharmacist professionalism	−0.303	0.167	0.739	0.531–1.024	−1.816	0.069
Good environment for privacy	0.408	0.241	1.567	0.957–2.573	1.697	0.074
Pharmacies adequately educate customers on patient privacy	−0.485	0.227	0.615	0.392–0.956	−2.142	0.032 *
Level of worry about privacy	0.107	0.168	1.114	0.802–1.552	0.642	0.520
Asked for unnecessary personal information	1.697	0.322	5.460	2.919–10.371	5.265	0.010 *
Avoiding discussion with pharmacist due to privacy concerns	1.456	0.396	4.291	2.038–9.740	3.675	<0.01 *
Health information treated confidentially	−0.202	0.353	0.817	0.403–1.620	−0.573	0.567

* *p*-value < 0.05.

## Data Availability

Data are contained within the article and Appendix A.

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
