# Peer review of "Privacy in Community Pharmacies in Saudi Arabia: A Cross-Sectional Study"

_healthcare, 2024, doi:10.3390/healthcare12171740_

Round 1

Reviewer 1 Report

Comments and Suggestions for Authors

Report of paper

Introduction: In the UK the GPhC requires all pharmacies to have consultation rooms- this information is specifically missing from this paper. Perhaps authors can specify this? 

Section 2 methodology - lines 82-92

line 104: Study design- did the authors validate their questionnaire?

Was a pilot study conducted –(authors state it was in line 104 – should it be in the methods section here?) Was a test-retest conducted for reliability? Was validity and reliability of the survey conducted and if so how?

line 114: authors state Cronbach’s alpha was used to test for internal consistency () -how? (if someone else wished to replicate your method how would they do this?)

line 115: what software was used to calculate the sample size?

When developing the questionnaire, was any literature used to compile the survey questionnaire?

Results: figs 1-3 are not very legible- need to increase font size for statements and numbers of percentages

line 216- discussion- it’s not only Australia but also UK requires pharmacies to have a consultation room or area;

Authors mention reliability under discussion- line 273- surely this should be addressed under results?  

Author Response

Dear Reviewer,

Thank you very much for your thoughtful and constructive feedback. I have carefully reviewed your comments and have provided detailed responses in the attached document. Your insights have been invaluable in improving the manuscript, and I greatly appreciate the time and effort you’ve invested in this review.

Best regards,

Reviewer 2 Report

Comments and Suggestions for Authors

General comments

This study on privacy practices in Saudi Arabia community pharmacies addresses a crucial healthcare issue. Its strengths lie in its relevance, adequate sample size (511 responses), and comprehensive approach to examining privacy perceptions and experiences. 

However, the research has several limitations:

  1. Lacks clear definitions of privacy concepts and international comparisons.
  2. The convenience sampling method via social media potentially impacts generalizability.
  3. Some statistically significant results have small effect sizes, requiring more discussion on practical significance.
  4. The sample demographics appear skewed, limiting representativeness.
  5. Data presentation could be improved for clarity.
  6. The use of odds ratios could have enhanced result interpretation.

Despite these limitations, the study provides valuable insights into pharmacy privacy concerns in Saudi Arabia. With methodological improvements and more nuanced result interpretation, this research could significantly contribute to enhancing privacy practices in pharmacy settings.

Technical observations:

  • on transparency: It would be advisable and useful to share the survey (translated) as supplementary material
  • on Introduction section: It would be beneficial to set the stage by explaining the concepts of privacy and data protection, as well as their relationship. While legislation in the USA (HIPAA) and the EU (GDPR) can be used as references, it's important to note their differences. Consider adding:
  • Definitions of privacy and data protection in healthcare contexts
  • Brief comparison of HIPAA and GDPR approaches, highlighting key differences in scope, application, and individual rights

on logistic regression analysis. Your expertise is crucial in interpreting results where the magnitude of beta is relatively small (e.g., 0.13 or similar). Although p<0.05 and statistically significant, the practical significance of these small effects should be discussed. Consider:

  • Discussing the practical implications of small beta values
  • Presenting standardized coefficients or effect sizes for better comparability
  • Using Odds Ratios (OR) alongside beta coefficients for easier interpretation

- on methodology. The authors noted that the statistical power was 80%, but they didn't discuss the quality of the samples. The convenience sampling through social media platforms could be considered a snowball sample. Please address:

  • How this sampling method might affect the claimed 80% statistical power
  • The impact on the generalizability of the results
  • Potential methods to improve sampling in future studies

- on limitations. Given the unlikely demographic data (e.g., 80% university-level education, 39% unemployment, average customer age of 33.5 years), consider:

  • Discussing the potential impact of this demographic skew on the results
  • Using post-stratification weighting techniques to adjust for these imbalances
  • Addressing how these demographics might limit generalizability
  • on presentation. Table 2 is too crowded and should be split into 3 tables.

Figures 1-3 are somewhat redundant (containing similar information to Table 1) and difficult to read. Consider alternative visualizations such as grouped bar charts or heatmaps.

- on logistical regression analysis. Consider using Odds Ratios (OR) instead of or alongside beta coefficients for a more straightforward interpretation of the results. Include confidence intervals for ORs to provide a measure of precision.

Author Response

(The authors gave the same response as above.)

Reviewer 3 Report

Comments and Suggestions for Authors

The paper is generally well-conducted and written, with clear objectives and an appropriate research design. However, some improvements could be made. 
1. Ensure the abstract accurately reflects the study's findings and conclusions. Sometimes, key points from the discussion or results are omitted in the abstract, which should be a concise summary of the entire paper.

2. The introduction offers a reasonable summary of the topic. However, it would be advantageous to provide a more comprehensive review of the literature. Consider including more recent studies or a broader range of references to strengthen the background.
3. Clearly state the gap in the existing literature that this study addresses.
4. Although the methodology is generally well-described, additional details on how the survey was developed, validated, and distributed is needed. 
5. The results are comprehensible, although several sections may be simplified to enhance reading. Consider restructuring the sections to enhance the coherent progression of information.
6. The discussion section should focus more on linking the findings to the literature. Addressing the study's limitations directly would also improve the conclusions.

7. Its recommended to add a detailed section of the findings' implications, including policy or practice recommendations.

Comments on the Quality of English Language

The English language is generally clear, but minor editing is needed to correct grammatical errors and improve sentence structure. 

Author Response

(The authors gave the same response as above.)

Round 2

Reviewer 2 Report

Comments and Suggestions for Authors

Dear authors,

Thank you for making significant changes to the initial version. I believe all topics raised were adequately addressed, and I recommend this paper for publication.

Author Response

Thank you for your time and effort you put to review our manuscript. Your comments have significantly improved the clarity and coherence of the manuscript.